# A Survey to Understand Parent/Caregiver and Children’s Views on Devices Used for the Administration of Oral Pediatric Medicines in Japan

**DOI:** 10.3390/children9020196

**Published:** 2022-02-03

**Authors:** Jumpei Saito, Hidefumi Nakamura, Jennifer Walsh, Akimasa Yamatani, Smita Salunke

**Affiliations:** 1Department of Pharmacy, National Center for Child Health and Development, Tokyo 157-8535, Japan; yamatani-a@ncchd.go.jp; 2Department of Research and Development Supervision, National Center for Child Health and Development, Tokyo 157-8535, Japan; nakamura-hd@ncchd.go.jp; 3Jenny Walsh Consulting Ltd., BioCity Nottingham, Pennyfoot Street, Nottingham NG1 1GF, UK; jenny@jennywalshconsulting.com; 4School of Pharmacy, University College London, 29-39 Brunswick Square, London WC1N 1AX, UK; s.salunke@ucl.ac.uk

**Keywords:** pediatric, oral medicines, administration, dosing, device, survey

## Abstract

Administration devices are crucial for the correct dosing of medicines to children. In countries outside Japan, oral droppers and syringes are reported to be preferred for the administration of oral liquid medicines to neonates and infants, whilst spoons and cups are more frequently used for older children. However, in Japan the majority of oral medicines are powders and the use of dosing devices in each pediatric age group is not well known. This study was performed as an observational anonymous questionnaire survey on devices for oral medicines in children aged 10 to less than 18 years and parents/caregivers on behalf of children aged from birth to less than 18 years. The results from 336 respondents showed that powders were most frequently prescribed in children aged less than 10 years old followed by liquids. Unlike previous reports, droppers were most frequently used in patients less than 12 months old, while household spoons were most frequently used in older children. Oral syringes were perceived as easy to use, which was in line with previous studies. Further cross-regional multi-countries study for establishment the guidelines on the choice of device will be needed.

## 1. Introduction

Administration devices are crucial for the correct dosing of medicines to children; however, handling and dosing errors commonly appear in daily practice [1,2,3]. The European Paediatric Formulation Initiative (EuPFI) consortium has previously recognized that there are gaps in the availability of pediatric medicine administration devices and challenges associated with their use [4,5]. In countries outside Japan, oral droppers and syringes are reported to be preferred for the administration of oral liquid medicines to neonates and infants [6,7], whilst spoons and cups are more frequently used for older children. However, in Japan, many oral medicines are solid oral dosage forms such as granules and powders, and the use of dosing devices for children in each age group is not well known.

The superiority of oral dosing syringes over measuring cups, measuring spoons and household spoons has been reported [8,9]; however, syringes are not always provided freely to outpatients in routine clinical practice in Japan. The type of drugs which are more likely to be used in the pediatric patients and preferred devices for them vary from country to country. The dispensing practice of prescribed medicines for children in Japan differs from that in other countries. For example, oral powder medicines are commonly dispensed as single-dose sachets at a pharmacy, and an administration device is not always provided to patients. Furthermore, measuring cups or droppers are provided with liquid medicines, which are also prescribed frequently for children [10]. However, it is unknown if these devices are suitable and are easy for parents, caregivers, and children to use.

This study is part of a larger study conducted by EuPFI to assess and compare the different devices used in different countries and seek feedback from parents/caregivers and children on the use of devices available to them. The aim of this pilot study was to perform an anonymous survey among parents/carers and children to understand what administration devices (e.g., spoons, syringes, droppers) are used to measure and give oral medicines to pediatric patients. and assess the ease of use of devices available to parents/caregivers and children in Japan. Additionally, the appropriateness of the choice of device for oral medicine administration has been evaluated, taking into consideration the results of a previous survey conducted outside of Japan [10,11,12,13].

## 2. Materials and Methods

### 2.1. Study Design

This study was an internet-based survey of parents/caregivers and children on usability of administration devices available in hospital setting in Japan.

#### 2.1.1. Study Population

This observational study on devices for oral medicines was carried out in the National Center for Child Health and Development (NCCHD) hospital, Tokyo, Japan. The subjects for this survey were children aged between 10 years and less than 18 years, and parents/carers on behalf of children aged from birth to less than 18 years.

#### 2.1.2. Composition of the Questionnaire

The EuPFI developed the survey which was piloted with a small sample (*n* = 57) of parents/caregivers and children in a school setting in UK. The feedback was incorporated before finalizing the survey for further distribution in different countries to assess and compare the different practices in administrating medicines, devices used, and ease of use of devices available in their country. The survey was published for pilot surveys in healthcare professionals (HCPs) and parents/caregivers/children [10]. The EuPFI survey was adapted and translated to Japanese language. Using this electronic self-administered anonymous questionnaire (Qualtrics Japan LLC, Tokyo, Japan), the demographic information of the participant (age, country, type of oral medicine, and type of device) and user-friendliness of the selected device were reported. The clarity of instructions for the selected device was also investigated. The questionnaire consisted of the following eight items to collate the required information: (#1) age of the child, (#2) relationship of the caregiver with the child who takes the oral medicine, (#3) dosage form of the most recently used oral medicine, (#4) devices used for oral medication administration, (#5) duration of the selected dosing device use, (#6) daily frequency of use of the selected device, (#7) user-friendliness of the device, and (#8) instructions on how to use the device (source and clarity of instructions). The detailed questionnaire is available as Appendix A.

### 2.2. Research Setting and Recruitment

The ‘Qualtrics^®^ (Qualtrics Japan LLC, Tokyo, Japan)’ application (automated web-based) was used to design the web-based questionnaire and to collect the survey responses. The web-form questionnaire was distributed via the LINE messenger application (LINE Co. Ltd., Tokyo, Japan) to parent(s) or caregiver(s) who visited the NCCHD hospital. In addition, the web-form survey could be accessed via the link on the NCCHD hospital website. The purpose and scope of the study were provided at the start of the survey on the first page of the web form. A survey period was set between May and August of 2021.

### 2.3. Analysis of the Data

The survey responses were compiled in an Excel sheet for detailed data analysis. Statistical tests were performed using the R software version 4.1.1. For nominal dependent variables, Fisher’s exact tests were used, where more than two variables were compared (effect of instruction providers on the friendliness of device use), the Marascuilo procedure, that enables us to simultaneously test the differences of all pairs of proportions when there are several populations under investigation, was used for a multiple comparison analysis [14].

### 2.4. Ethics Approval

The consent and assent forms were embedded in the survey, and consent or assent was obtained before participants were permitted to start the survey. This study was reviewed and approved by the NCCHD Research Ethics Committee (“Protocol # NCCHD2021-008” as per the updated guidelines of the Declaration of Helsinki (64th WMA General Assembly, Fortaleza, Brazil, October 2013) and of the Policy for the Protection of Human Subjects) on 30 September 2021 (Appendix A).

## 3. Results

### 3.1. Participants

In this study, 580 survey evaluations were collected, of which 578 (99.7%) were from Japanese respondents and two were from Americans. The flow chart of data collection for the analysis is shown in Figure 1. Of all respondents, 434 completed all questionnaire items; 341 (78.6%) were from parents, 70 (16.1%) were from children aged between 10 years and less than 18 years, and the remaining 23 (5.3%) were from other caregivers (grandparents or other adult family members). The number of parent-child pairs was 70. Regarding the pediatric age group, 34 (7.8%) evaluations were for children aged less than 12 months, 64 (14.7%) were for children between 12 and 23 months, 188 (43.3%) were for children aged 2 to 5 years old, 68 (15.7%) were children aged 6 to 9 years old, and 80 (18.4%) were children aged 10 to less than 18 years old. In total, 336 respondents used at least one device for taking oral medicine.

### 3.2. Dosage Form of Oral Medicine the Child Had Taken

The type and frequency of administrated dosage forms in each age group are shown in Figure 2. In patients less than 10 years old (*n* = 354), powders including fine granules, granules, and dry syrups were most frequently prescribed (*n* = 252, 71.2%). Patients aged less than 6 years were also most frequently prescribed powders, with liquids including syrups and suspensions being prescribed the second most frequently (*n* = 60, 21.0%). The frequency of liquid preparation decreased with increasing age, whilst the prescription frequency for tablets increased with age, with 72 (48.6%) of all children aged between 6 years and less than 18 years old being prescribed this dosage form. The two tablet prescriptions for children aged less than 12 months were orodispersible tablets.

### 3.3. Type of Device for Medicine Administration

Figure 3 shows the frequency of devices used in each age group. Droppers were the most frequently used device in children aged less than 12 months (*n* = 14, 41.2%). The second and third most used devices were household spoons (*n* = 6, 17.6%) and syringes (*n* = 4, 11.8%), respectively. In patients aged 1 to 5 years old (*n* = 252), household spoons were most frequently used (*n* = 84, 33.3%), followed by syringes, droppers, and measuring cups. In patients aged 6 to 9 years old (*n* = 68), measuring cups were most frequently used (*n* = 18, 26.5%), followed by household spoons and syringes. In children older than 10 years (*n* = 80), household spoons were most frequently used (*n* = 14, 17.5%); however, half of the children older than 10 years take their medicine without using a device, and reporting rates of “no device was used” increased with age.

Appendix A shows devices children or caregivers used for taking each type of medicine. More than half of the respondents who took tablets or liquids did not use any devices. For taking powders, various devices were used, household spoons being the most frequently used. For liquids, a measuring cup was most commonly used followed by droppers and syringes.

### 3.4. Duration of Device Use

The duration and frequencies of each device use are indicated in Table 1. Of all 336 respondents who used at least one device for taking an oral medicine, 243 (72.3%) had used the selected device for more than 1 month, 46 had used it for 1 to 11 months, and 197 had used it for more than 1 year. Regarding the frequency of the device use, 86 (25.6%) and 183 (54.5%) of respondents had used the selected device once and twice daily respectively. These frequencies of use accounted for 80.0% of all respondents who had used a device for oral medication.

### 3.5. User-Friendliness of the Selected Device

All 336 participants answered the question on the user-friendliness of the selected device. The number of respondents selected “easy to use” and “difficult to use/neither” were 92 and 24 for household spoons, 60 and 2 for syringes, 46 and 12 for measuring cups, 42 and 10 for droppers, 14 and 4 for other devices including straw, wafer sheet, and nipple. The ratio of “easy to use” and “difficult to use/neither” was 3.8, 30.0, 3.8, 4.2, and 10.0, for household spoons, syringes, measuring cups, droppers, and other devices. respectively (Figure 4). Statistical analysis suggests that syringes were considered easy to use compared to the other types of devices (*p* < 0.01, Fischer’s extract test).

Regarding the user-friendliness of each device in each age group (Appendix A), overall, most devices were recognized as “easy to use”; however, the number of “easy to use” responses did not always exceed “difficult to use” responses for some devices in specific age groups. For example, no remarkable difference was observed between “easy to use” and “difficult to use/neither” for measuring cups and household spoons in children aged less than 12 months. The same applied to cups for children aged 12 to 23 months, household spoons for children aged 6 to 9 years old, and droppers for children aged 10 to less than 18 years old.

Focusing on specific combinations between device and formulation type, the use of household spoons for taking liquids appeared to be difficult since the “difficult to use/neither” responses outnumbered the “easy to use” responses, with the majority of “difficult to use” responses being from parents/caregivers whose children were aged 12 to 23 months (Appendix A). In contrast, the administration of powders with household spoons appeared to be acceptable (Appendix A).

Due to the limited number of subjects, the effect of age on the user-friendliness of each device in every type of formulation was not analyzed.

### 3.6. Clarity of the Instruction for Devices

Of all the respondents who used a device for oral medication, 202 used the provided device from a HCPs at a pharmacy or a hospital. Of them, 107 (53.0%) respondents indicated some instructions on how to use the device were provided. The question regarding source of information permitted multiple answers, and a total of 143 answers were collected. Respondents stated that instructions were provided from a pharmacist (*n* = 63, 44.1%), a physician (*n* = 33, 23.1%), nursing staff (*n* = 31, 21.7%), and others (not specified, *n* = 7, 4.9%). The patient information leaflet was also referred to as a source of information (*n* = 9, 6.3%). Regarding the clarity of provided instructions for the devices, 90 (84.1%) respondents reported the provided instructions were clear, whilst one respondent (0.9%) indicated instructions were not clear, and 16 (15.0%) reported they were neither clear nor not.

Statistical analysis suggested that the provision of instructions, clarity of instruction, and source of instruction providers (pharmacist, nursing staff, physician, or patient information leaflet) use did not affect the user-friendliness of device use (*p* > 0.05, Fisher’ exact tests and multiple comparisons with Marascuilo procedures).

## 4. Discussion

This study was conducted to understand what administration devices are used to measure and give oral medicines to pediatric patients and assess the ease of use of devices available to parents/caregivers and children in Japan.

The results of this survey have provided some valuable insights regarding the use of oral pediatric medicine administration devices in Japan. Powders were most frequently prescribed to children less than 10 years old and were commonly administered using a dropper in patients less than 12 months old and using a household spoon in those aged 12 months to less than 6 years; in Japan, powders are dispensed in liquid form before being administered to the patient [15]. The previous HCPs study conducted in six European countries indicated that dosing cups were frequently used in older children [10], a finding that differed from that of our study. The European study reported that oral liquids were commonly prescribed to children [16], and the differences in dosage forms prescribed between Japan and other countries, and method of supply may affect device selection. Measuring cups and household spoons, which are recognized as “traditional devices” [17] were frequently used for liquid administration in Japan, whereas oral syringes (dispensers) which may be considered to be “innovative devices” were frequently used for oral liquids in European countries (Albania, Italy, the Netherlands, Romania, Spain, UK), Israel, and the United States of America [13]. The trend for traditional device use for oral liquids in Japan was similar to results reported from a pan-Indian survey [11].

Although oral syringes are considered to provide more accurate dosing than cups and spoons, the risk of inaccurate dosing of liquids when using these so-called “traditional devices” may not be recognized in Japan, and hence their frequent use. Furthermore, the proportion of oral liquid prescriptions compared to powders is small in Japan.

For powdered drugs, a household spoon was most frequently used in Japan. Spoons are considered to give incorrect medicine doses [18]; however, in Japan, powdered drugs are dispensed and administered as single-dose sachets. Dispensing as single-dose sachets may contribute to a low risk for dosing errors since no measurement of the dose is required by the caregiver. This is an insight that may support better-dosing accuracy levels in a home situation and therefore may be a more universally suitable method for presentations of children’s medicine. Regarding the user-friendliness of dosing devices, a preference for oral syringes was higher than other devices, which is in line with a previous study [9]. The instruction of device usage from an HCP did not appear to have an impact on the user-friendliness of each device.

From the current prescription status of oral medicines for pediatric patients in this study, the primary type of medicine is powders. Since most powder medicines are packaged in single-dose units, caregiver does not need to measure the required dose. Hence, the risk of dosing errors may be considered low, as long as the entire sachet contents are taken. However, liquids were the second most prescribed formulation in patients less than 6 years old and were administered with droppers and household spoons as well as syringes. Whether pharmacists or healthcare professionals in Japan are aware of the risk of dosing errors for taking liquid medicine with spoons or droppers is not known. Confirmation of the accuracy of dosing liquid medications using the provided devices in patients in Japan is required.

As the limitation of this study, the number of responses was limited to find the appropriate device in each age group to suggest a suitable device for each type of medicine by age, a large and multicenter study will be required.

This study identified several differences between practices in Japan and other countries regarding oral medicine dosage form and device used, administration device usability from the parents, caregivers, and children’s perspectives, and the frequency of devices used. Further cross-regional studies would be valuable to assist with establishment of guidelines that reflect the circumstances of each country.

## 5. Conclusions

Powders were most frequently prescribed in children aged less than 10 years old, and droppers were most frequently used in patients less than 12 months old, while household spoons were most frequently used in older children. Oral syringes were perceived as easy to use, which was in line with previous studies. Further cross-regional multi-countries study for establishment the guidelines on the choice of device will be needed.

## Figures and Tables

**Figure 1 children-09-00196-f001:**
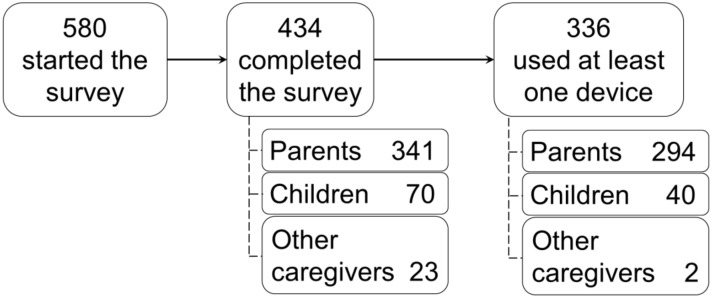
Flow chart of data collection.

**Figure 2 children-09-00196-f002:**
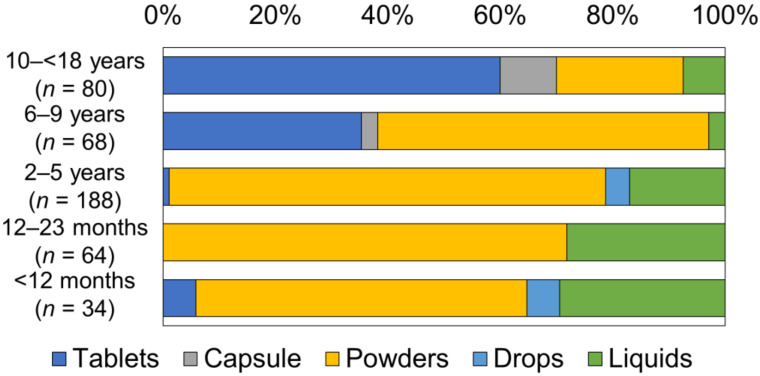
The frequencies of prescribed dosage forms in each age group.

**Figure 3 children-09-00196-f003:**
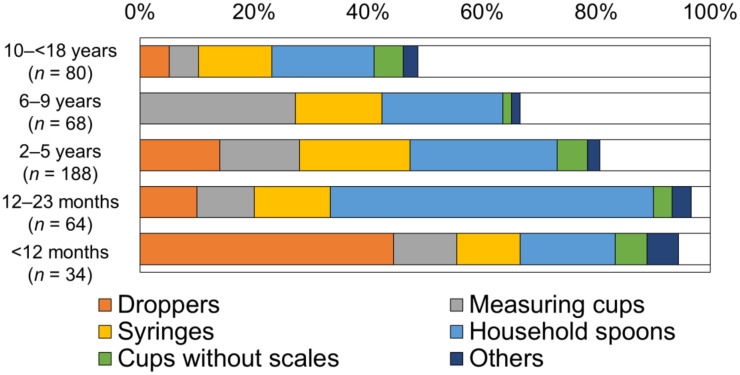
The frequencies of devices used in each age group.

**Figure 4 children-09-00196-f004:**
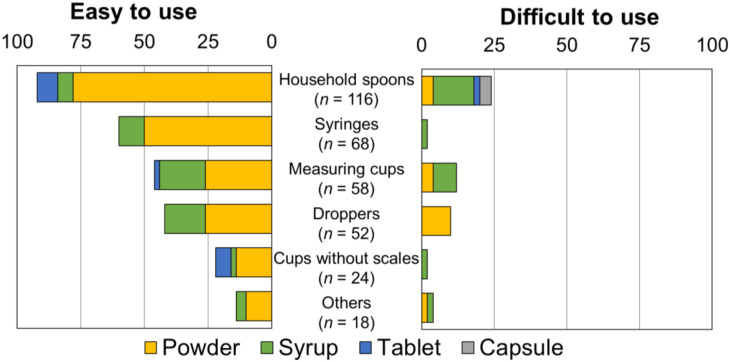
The user-friendliness of used devices.

**Table 1 children-09-00196-t001:** The duration and frequencies of each device use.

Duration	N (%)		Daily Frequencies	N (%)	
>7 days	35	(10.4)	Once-daily	86	(25.6)
1 to 2 weeks	32	(9.5)	Twice daily	183	(54.5)
3 to 4 weeks	18	(5.4)	Three times a day	50	(14.9)
1 to 11 months	46	(13.7)	Four times a day	7	(2.1)
>1 years	197	(58.6)	Others	10	(3.0)
Unknown	8	(2.4)			
Total	336		Total	336	

## Data Availability

Not applicable.

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
