# Peer review of "A Survey to Understand Parent/Caregiver and Children’s Views on Devices Used for the Administration of Oral Pediatric Medicines in Japan"

_children, 2022, doi:10.3390/children9020196_

Round 1

Reviewer 1 Report

Comments to authors

General

This manuscript presents information which is complementary to the work of the European Pediatric Formulation Initiative and serves to highlight differences in pediatric medicine’s practice in Japan, which will be of interest to researchers working in this area. The manuscript is relatively well written and clear, although there are numerous places where wording, grammar and syntax should be improved as noted below.

Specific comments

1) The title should probably specify that the subject is administration of oral pediatric medicines, since various devices are used with other routes of administration. The current mss contains no comments, for example, on use of inhalers.  

2) Abstract, lines 26-8

Sentence is incomplete and should be rewritten.

3) Introduction, line 38

Meaning is unclear; perhaps ‘and’ should be deleted.

4) Introduction, lines 44-5

Unclear intent…should be rewritten.

5) Introduction, line 60 : Reference appears to be made to at least two previous surveys, although it is possible that the Indian survey is the only comparator

6) Results, lines 127-8

Meaning is not clear.

7) Section 3.4, line 170 : Reference to ‘either device’ is confusing since there at least five possibilities.

8) Section 3.5, line 187

Reference is made to Figure 4 but this was not included with the review material.

9) Section 3.5, line 208 and elsewhere : Reference is made to HCPs. Presumably this means health care practitioners, or providers, or professionals. Meaning should be specified at least once.

10) Section 3.5, line 219 :Table 2 is repetitive with the text and adds no new information.

11) Discussion, line 227 : Meaning unclear; should be rewritten.

12) Discussion, line 230 : ‘Provided’ would be a better word; insights were not "identified."

13) Discussion. line 237

‘ …which differed from our results’ should be rewritten as ‘a finding that differed from that of our study’.

14) Discussion, line 262 : Suggest deleting ‘parents’, so that subject and verb agree.

15) Discussion, line 263 : Suggest replacing ‘whole’ with ‘entire’.

16) Discussion, line 264 :Suggest adding ‘formulation’ after ‘most prescribed’.

17) Discussion, lines 267-8

‘Confirming and recognizing’ could be replaced by ‘confirmation of’.

18) Discussion, lines 270-1 : Confusing syntax, should be rewritten.

19) Conclusion, lines 283-4 : Suggest rewriting ‘further cross-regional studies would be valuable to assist with establishment of guidelines…’

20) References

Reference 7, line 331 is incomplete.

Author Response

Thank you very much for providing important insights. We are delighted to hear that you think our work will spark debate in our field. In the following sections, you will find our responses to each of your points and suggestions. We are grateful for the time and energy you expended on our behalf.

The response to the reviewer's comments was prepared and attached as a separate word file.

Reviewer 2 Report

Dear Author,

The research article entitled “A survey to understand parent/caregiver and children’s views on devices used for the administration of pediatric medicines in Japan” has been intensively reviewed and evaluated. This endemic survey study has been considered as original research after the inspection of academic databases.

Hereby, I would like to present my tiny suggestion and one revision. Would authors please respond to the revisions or suggestions to enrich the contents of their work?

  1. Revision_1: Could authors explain the selection of statistical method used in their study which was mentioned “Analysis of the data” section at the beginning of the “discussion” section (briefly).
  2. Suggestion_1: Authors explain all the supplementary data clearly, as a suggestion it could be beneficial to add an ethics approval form that is provided by the NCCHD Research Ethics Committee.

Author Response

Thank you very much for providing important insights. 

The response to the reviewer was prepared as a separate file.

Thank you again for taking your time.
